# Skeletal Muscle Mass, Sarcopenia and Rehabilitation Outcomes in Post-Acute COVID-19 Patients

**DOI:** 10.3390/jcm10235623

**Published:** 2021-11-29

**Authors:** Michele Gobbi, Emanuela Bezzoli, Francesco Ismelli, Giulia Trotti, Stefano Cortellezzi, Francesca Meneguzzo, Marco Arreghini, Ionathan Seitanidis, Amelia Brunani, Valentina Aspesi, Veronica Cimolin, Paolo Fanari, Paolo Capodaglio

**Affiliations:** 1Orthopaedic Rehabilitation Unit and Research Laboratory in Biomechanics and Rehabilitation, Istituto Auxologico Italiano, IRCCS, 28824 Piancavallo, Italy; francesco.ismelli@gmail.com (F.I.); giuliatrotti@gmail.com (G.T.); cortellezzistefano@gmail.com (S.C.); meneguzzo.francesca93@gmail.com (F.M.); m.arreghini@auxologico.it (M.A.); i.seitanidis@auxologico.it (I.S.); brunani@auxologico.it (A.B.); v.aspesi@auxologico.it (V.A.); p.capodaglio@auxologico.it (P.C.); 2Respiratory Rehabilitation, Istituto Auxologico Italiano, IRCCS, 28824 Piancavallo, Italy; e.bezzoli@auxologico.it (E.B.); p.fanari@auxologico.it (P.F.); 3Department of Electronics, Information and Bioengineering, Politecnico di Milano, 20133 Milano, Italy; veronica.cimolin@polimi.it; 4Department of Surgical Sciences, Physical Medicine and Rehabilitation, University of Torino, 10126 Torino, Italy

**Keywords:** rehabilitation, SARS-CoV-2 infection, COVID-19, sarcopenia, nutritional status, functional evaluation

## Abstract

The relationship between skeletal muscle mass at the beginning of the post-acute rehabilitation phase and rehabilitation outcomes has been scarcely investigated. The aim of this study was to investigate the impact of the existence of sarcopenia upon admission to a post-acute COVID-19 patient rehabilitation unit on body composition and functional and respiratory capacity at discharge. Thirty-four post-acute COVID-19 patients were referred to our Rehabilitation Unit from different COVID Hospitals in northern Italy. Body weight loss, body composition, handgrip strength, functional parameters, oxygen saturation and related perception of dyspnea in several positions were measured before and after a 28-day multidisciplinary rehabilitation program. Spirometry was performed only upon admission. The intervention included psychiatric support, cognitive behavioral therapy, nutritional therapy and physiotherapy, including aerobic and resistance training. Training volume was 45 min/session, 6 sessions/week. Upon admission, the prevalence of sarcopenia among our patients was 58%. In all of the 34 patients, we observed a trend of improvement in all of the respiratory, body composition, muscle strength and functional parameters considered. Monitoring muscle mass and strength in post-acute COVID-19 patients appears to be a key predictor of rehabilitation outcomes. Early diagnosis of sarcopenia therefore appears to be of paramount importance in the management of post-acute COVID-19 patients.

## 1. Introduction

COVID-19 is an infectious disease caused by SARS-CoV-2, leading to a severe acute respiratory syndrome and multi-organ failure [1]. COVID-19 patients may develop acute respiratory, nervous and musculoskeletal symptoms, leading to difficulty in verticalization [2,3]. In the subacute phase, COVID-19 patients require intensive rehabilitation aimed at restoring independence in basic daily activities. Some patients may develop post-viral fatigue syndrome or “long COVID” [3]. Rehabilitation guidelines for post-COVID-19 patients have been released by the World Health Organization [4] and different countries [5]. To update the rehabilitation community on the growing evidence for the role of rehabilitation in management of COVID-19 patients, Cochrane Rehabilitation launched REH-COVER [6]. Early mobilization with frequent posture changes, bed mobility, sit-to-stand, simple bed exercises, while respecting the patient’s respiratory and hemodynamic status [2], and active limb exercises accompanied by progressive muscle strengthening [7] are recommended. Progressive aerobic exercise should then be increased to 20–30 min daily [8]. In COVID-19 patients, malnutrition is frequently observed [9], because of a direct effect of the virus and the frailty of these patients due to the presence of other infections or systemic organ failure [10], gastro-intestinal symptoms, long hospitalization and ICU stay with prolonged immobilization [11]. Altogether, impaired food and protein intake and absorption, inflammation, low vitamin D levels and presence of acute conditions could lead to anabolic resistance and acute sarcopenia, a metabolic condition with a reduced muscle protein synthesis, also favored by factors such as aging, frailty and obesity [12]. Malnutrition is a strong predictor of sarcopenia [13] and indeed, survivors of COVID-19 are at increased risk of acute sarcopenia [12] due to loss of muscle mass, fiber denervation, neuromuscular junction damage and upregulation of protein breakdown [14]. The relationship between skeletal muscle mass at the beginning of the post-acute rehabilitation phase and rehabilitation outcomes has been scarcely investigated, and a deeper understanding would contribute to increased effectiveness of rehabilitation programs. The aim of this study was to investigate the impact of the existence of sarcopenia upon admission to a post-acute COVID-19 patient rehabilitation unit on body composition and functional and respiratory capacity at discharge.

## 2. Materials and Methods

From April to June 2020, we recruited 34 post-acute COVID-19 patients re-referred to our Rehabilitation Unit from several COVID hospitals in the north of Italy, consecutively. All patients presented a history of COVID-19 disease documented by real-time PCR and next-generation sequencing positive swabs, and with different clinical manifestations. Before admission to our Rehabilitation Unit, patients had to present two consecutive negative swabs for SARS-CoV-2. This study was conducted according to the Declaration of Helsinki of the World Medical Association and was approved by the Ethics Committee of the Istituto Auxologico Italiano (2020_05_19_04). Written informed consent was obtained from all experimental patients.

### 2.1. Measurements

Body height (to the nearest 0.1 cm) and weight (to the nearest 0.1 kg) were measured in light indoor clothing. Body mass index (kg/m^2^) was calculated. Upon admission, individual weight loss was calculated from the body weight value prior to the COVID-19 disease, held as “habitual body weight”. Body composition was assessed in the morning in a comfortable room at a temperature of 22–25 °C by Bioelectrical Impedance Analysis (BIA 101/s, Akern^®^, Florence, Italy) with the patient in a supine position, with the lower limbs slightly separated and an empty bladder. Whole-body resistance (Rz) and reactance (Xc) were measured by expert operators with an impedance analyzer that delivered a 50 kHz sinusoidal alternating current connected to four surface electrodes placed on the right foot and hand [8]. To assess the Appendicular Skeletal Muscle Mass (ASM), we used the two following formulas [9]:ASM beyond 65 years = −4.211 + (0.267 × height^2^/Rz) + (0.095 × weight) + (1.909 × gender (male = 1, female = 0)) + (0.012 × age) + (0.058 × Xc) 
ASM over 65 years = 3.964 (0.227 × RI) + (0.095 × weight) + (1.384 × gender) + (0.064 × Xc),
where RI indicates Rz normalized for height [15]. The cut-off values from the European Working Group on Sarcopenia in Older People (EWGSOP2) [16] were used to diagnose sarcopenia: ASM < 20 (kg) for males and ASM < 15 (kg) for females. We also assessed the phase angle (PhA), a derived measure obtained from the equation: phase angle (degrees) = arctan(Xc/R) × (180/π) [17]. Despite variability in PhA due to gender (higher values in males than females), aging and BMI > 40 kg/m^2^ [18], it is known that low PhA values (<5) are associated with poor nutritional status and outcome [17]. We measured muscle strength and performance scores with handgrip (HG) and time-up-and-go (TUG) tests, respectively. We utilized a Jamar analog dynamometer (Asimow Engineering Co., Los Angeles, CA, USA) to measure force. The average value of three repetitions of grip strength for each hand was used and compared with current normative values for males (>27 kg) and females (>16 kg) [16]. Physical performance was measured with the TUG test [19]: a TUG value greater than 13.5 s is considered a practical cut-off for high fall risk [20], while a 20 s value indicates severe sarcopenia [16]. At baseline, 34 (M/F 16/18) post-COVID-19 patients were assessed. We divided patients into two subgroups (S—sarcopenic group, NS—non-sarcopenic group) based on the diagnosis of sarcopenia considering the cut-offs defined by Cruz-Jentoft et al. [16]. Oxygen saturation and related perception of dyspnea were monitored in lying, sitting and standing positions and after completion of TUG. A spirometry test was performed upon admission to the rehabilitation unit and was not repeated at discharge, due to limited access to respiratory testing because of COVID restrictions.

### 2.2. Intervention

Consistent with current consensus statements for post-COVID-19 rehabilitation (4–8), our multi-disciplinary rehabilitation program is intended to help recover from functional impairment and limitations of the respiratory, cardiac, musculoskeletal and neurological systems. It includes psychiatric support, cognitive behavioral therapy to reduce the post-traumatic stress symptoms and nutritional treatment to improve malnutrition. All patients recruited were from other hospitals in the same or neighboring regions. Based on the medical records, 11 of the 34 patients were hospitalized in the ICU. Unfortunately, we are not aware of the number of days spent in the ICU because they were not always reported. In our sample, prevalence was 17.1% for hypertension, 16.4% for cardiac-cerebrovascular disease and 9.7% for diabetes. Length of the program was on average 28 days. Each patient was individually trained by a physical therapist to initially perform sitting-to-bed exercises for upper-body conditioning [2,8] and progressive limb muscle strengthening (8–12 repetitions, 1–3 sets, with a 2 min rest between sets). After several days, progressive aerobic exercises with a cicloergometer and armergometer were introduced. The latter were performed at moderate intensity, i.e., 65% of maximal heart rate according to the equation ((220 − age) × 0.65). This approach was individualized according to the patient’s conditioning, subjective perception of fatigue and clinical status. The goal of training volume was 45 min/session, 6 sessions/week. Training was tracked with subjective perception of breathlessness according to the modified Borg scale (MBS) [21] and peripheral blood oxygen level with a pulse oximeter (Nonin; Palm Sat 2500a, RH Tilburg, The Netherlands). The latter was measured in different (supine, sitting and standing) positions and after the execution of TUG, with the aim of detecting any differences in respiratory capacity secondary to a common daily task [22]. The clinical significance of MBS has been validated before [23]. Baseline, post-exertion oxygen saturation and post-exertion change in oxygen saturation (baseline—post-exertion oxygen saturation) were monitored.

### 2.3. Nutritional Intervention

The nutritional intervention was based on the recent practical guidelines established by the European Society for Clinical Nutrition and Metabolism (ESPEN) for the nutritional management of COVID-19 patients [1]. According to the expert panel, total energy intake was calculated in the range of 27 to 30 kcal/die/kg considering the adjusted body weight (AjBW = ideal body weight ((actual body weight/ideal body weight) × 0.33)) [1] for BMI > 28 kg/m^2^ or usual body weight for BMI < 28 kg/m^2^ before COVID-19 disease. Likewise, we calculated protein intake in the range of 1 to 1.3 g/die/kg body weight. An ideal energy ratio of fat to carbohydrate was 30:70, or 50:50 in patients supplemented with O_2_. In an additional way, as recommended by ESPEN guidelines, patients were supplemented with oral multi-minerals (K, Ca, P, Mg, Mn, MO, Se, Cr, I, Fe, Zn, Cu) and multivitamins (vitamin A, D, E, K, C and B vitamins) (Sypradyn refill; Bayer S.p.A., Garbagnate Milanese (MI); Italy), and vitamin D (1000 IU/day, 10,000 IU/week, 25,000 IU/2 weeks, 50,000 IU/4 weeks) and essential amino acids (4 to 8 g/day).

### 2.4. Outcomes

Clinical examination, anthropometry, laboratory findings, body composition (ASM and PhA), hand grip strength (HG), performance test (TUG), oxygen saturation and related perception of dyspnea in the lying, sitting and standing positions and after completion of TUG were performed at T0 and T1 to evaluate the effectiveness of the rehabilitation program. Spirometric testing was performed only at T0 to evaluate the baseline pulmonary capacity of patients.

### 2.5. Statistical Analysis

Continuous variables were expressed as median and interquartile range. Differences between groups were assessed using Fisher’s test for continuous variables, or the c2 test for categorical data, as appropriate. A *p*-value < 0.05 was considered to be statistically significant. Statistical analyses were performed with SPSS 25.0 (IBM, Armonk, New York, NY, USA).

## 3. Results

Twenty patients (M/F 11/9, mean age 71.5 (17.0) years, mean BMI 21.0 (4.2) kg/m^2^) were classified as sarcopenic. The number of patients admitted to the intensive care unit in the previous hospitalization was 8/20 based on the clinical records. Fourteen patients (M/F 5/9, mean age 68.0 (16.5) years, mean BMI 28.4 (8.9) kg/m^2^) were not sarcopenic. The number of patients admitted to the intensive care unit in the previous hospitalization was 3/14.

In our sample, comorbidity prevalence was 17.1% for hypertension, 16.4% for cardiac-cerebrovascular disease and 9.7% for diabetes. Anthropometric, body composition, muscle strength and functional values of both subgroups at admission (T0) and discharge (T1) are reported in Table 1. At T0, S and NS subgroups presented a different picture: NS had a younger mean age and a higher BMI than S, both S and NS presented a reduced phase angle (<5°), a proxy of reduced nutritional status, but ASM, handgrip strength and TUG were reduced only in S (Table 1). At T1, both S and NS improved TUG and ASM, PhA improved only in S and handgrip strength improved only in S, with only female patients reaching significant changes (Table 1). Additionally, the need for walking aids was reduced at T1. At T0, in S, 4 patients needed 4-wheel walkers, 3 patients needed 2-wheel walkers, 1 patient needed 2 canes and 1 patient needed 1 cane; in NS, 2 patients needed 4-wheel walkers and 1 patient need a cane, while at T1, only 2 patients needed aids and only 6 patients were still using a single cane.

Respiratory parameters in S and NS at T0 and T1 are reported in Table 2. In S, almost all parameters analyzed showed a statistically significant improvement at T1. In NS, only SpO_2_ after TUG showed a statistically significant improvement at T1. Spirometry was evaluated only at T0: we found a FEV1 of 1.98 L (SD: 0.70 L) and a FVC of 2.30 L (SD: 0.81 L) in S, and a FEV1 of 2.16 L (SD: 0.73 L) and an FVC of 2.58 L (SD: 0.88 L) in NS. In S, the correlation between ASM and FVC and FEV1 was statistically significant (0.582 and 0.592, respectively), whereas in NS, the correlation was not significant. Main laboratory findings at T0 and T1 are reported in Table 3. At T0, albumin and vitamin D levels were low [24] in S. In NS, albumin, ferritin and vitamin D levels were higher than S, however the level of vitamin D was still insufficient [24]. At T1, ferritin and vitamin D showed a statistically significant improvement in S, whereas in NS a statistically significant improvement could only be seen in ferritin level.

## 4. Discussion

Upon admission to the Rehabilitation Unit after the acute phase, we documented a 58% prevalence of sarcopenia among our 34 post-COVID-19 patients, according to the cut-off values by Cruz-Jentoff et al. This is in line with recent reports on the prevalence of sarcopenia in intensive care units [25]. Known factors that may have contributed to a state of sarcopenia were: malnutrition, increased protein and energy requirements, weight loss, prolonged immobilization, low-calorie and low-protein diet, low vitamin D level, inflammatory state, BMI and age. A risk of malnutrition in post-acute COVID-19 patients has been previously documented [9]. Indeed, malnutrition seems to increase the risk of sarcopenia by 13 times [26]. Jeon et al. reported that height-adjusted ASM showed a positive correlation with FEV1 and FVC, suggesting that a reduction in pulmonary function may parallel a loss in muscle mass [27]. It has been suggested that sarcopenic subjects have a thinner diaphragm than their non-sarcopenic counterparts, which is associated with reduced respiratory function and lower ASM [28]. The decline in muscle strength therefore appears not to be limited to limb muscles [28]. Furthermore, according to the Chinese Clinical guideline for the classification of COVID-19 severity, our sarcopenic patients were ranked as severe due to the presence of an oxygen saturation ≤ 93% at rest [29]. Our data at T0 show lower circulating levels of vitamin D (13.7 (13.2) vs. 25.5 (13.5)), higher inflammatory state ferritin in S (448.5 (367.0)) vs. ferritin in NS patients (151.0 (197.0)) and more advanced age (71.5 (17.0) vs. 68.0 (16.5)) in S as compared to NS patients. The literature supports the role of hypo-vitamin D in reducing muscle protein synthesis, growth and strength [26,30]. A supplementation of vitamin D could lead to an increase of the cross-sectional area of type 2 fibers [30]. Age and severe inflammation caused by COVID-19 [2] can cause a loss of lean mass and a reduction in strength [30]. Vitamin D also has anti-inflammatory properties [30]. In the presence of an underlying illness (either acute or chronic), inflammatory background is responsible for a highly catabolic state [2], negatively affecting energy balance and resulting in significant but not exclusive lean body mass breakdown [30]. Indeed, a loss in lean mass was documented with CT scans in post-COVID-19 obese and non-obese patients after lengthy hospitalization or ICU [31]. In our study, a BMI > 30 in NS patients may have played a protective role with regard to the development of sarcopenia. It is known that BMI is positively associated with ASM and muscle strength [30]. Our results showed a trend of improvement in all of the respiratory, body composition, muscle strength and functional parameters in both S and NS groups, considered after a comprehensive 28-day rehabilitation program. Sarcopenic patients showed more pronounced, statistically significant improvements at T1 in all domains (respiration, body composition, muscle strength and function) as compared to NS patients. At T1, parameters of strength, body composition and functional capacity improved. ASM in S (*p* = 0.001) and NS (*p* = 0.009) groups showed a statistically significant improvement only in female patients. HG improved in S patients, but the improvement was only significant in females (*p* = 0.035 *). TUG improved significantly in both S (*p* = 0.039 *) and NS (*p* = 0.045 *) groups. A predominance of different muscle fibers and a different cross-sectional area could account for gender differences, as type 1 fibers are predominant in females. The improvement in ASM observed only in female S patients may be attributable to such gender differences. Increased muscle mass may require more dietary protein in order to achieve a positive net protein balance and overcome muscle protein breakdown [32]. It should always be considered that the BIA methodology for assessing body composition is affected by variations in total body water. Improvements in strength, body composition and functional performance could be expected from a resistance training program and it is possible that vitamin D supplementation may have modulated muscle growth and strength gains [33]. Additionally, it can be speculated that patients in the S group, starting from a more severely de-conditioned status, benefited more from the rehabilitation intervention than NS patients. It is known that the relationship between strength and function is curvilinear, suggesting that factors other than muscle function contribute to function; still, muscle strength is considered the most important [34]. Hand grip strength is a proxy for global strength measurements, and it has been used to identify sarcopenia in clinical and community health settings [35]. Deconditioned, frail patients below the threshold of independence more likely achieve clinically significant functional gain in basic activities of daily living, as compared to patients who are already functionally closer to independence, namely the NS patients. At T1, all respiratory parameters improved in S patients. The relationship between increased muscle mass and muscle strength in relation to respiratory muscles has been recently investigated: respiratory muscles, especially inspiratory muscles, seem to be significantly related to limb muscle strength and skeletal muscle mass [36]. It can be speculated that the improvements in muscle and body composition and inflammatory and functional status led to an improvement in respiratory parameters. At T1, vitamin D and ferritin significantly improved in S patients. This improvement supports a better anabolic and inflammatory status that is involved in a reduced catabolic state. The majority of studies investigating sarcopenia in humans have suggested that loss of muscle mass is primarily driven by a blunted synthetic response to both feeding and exercise, identified as anabolic resistance [37]. Muscle protein loss can occur as a result of increased muscle protein breakdown or reduced muscle protein synthesis (or by some combination of the two). It has been well-established that a prolonged period of disuse can lead to a rapid loss of total skeletal muscle mass and strength. The combination of therapeutic exercise, aerobic exercise and resistance training, and a higher protein diet with the use of dietary supplements of vitamin D and essential amino acids, leads to a reduction in the inflammatory state of the patient by restoring the balance between anabolic and catabolic states, counteracting the muscle protein breakdown and the anabolic resistance [38]. Based on the collected data, monitoring muscle mass and strength in post-acute COVID-19 patients could be useful to predict respiratory, functional and nutritional rehabilitation outcomes.

## 5. Conclusions

We documented a 58% prevalence of sarcopenia among our 34 post-COVID-19 patients. This sample is small and further investigation is necessary. Early diagnosis of sarcopenia appeared to be of paramount importance in the management of post-COVID-19 patients to improve their rehabilitation outcomes and long-term prognosis [1]. SARS-CoV-2 infection together with intensive care and related complications can lead to acute sarcopenia even in younger patients. Rehabilitation, and in particular resistance training, in those patients is crucial for recovering independence, and rehabilitation protocols should be tailored to the individual clinical status and integrated with appropriate nutritional interventions.

## Figures and Tables

**Table 1 jcm-10-05623-t001:** ICU stay in previous hospitalization, anthropometric, body composition (PhA, ASM), muscle strength (handgrip) and functional (TUG) parameters at T0 and T1 in both the sarcopenia group (S) and non-sarcopenia group (NS) for males (M) and females (F). All data are reported as mean ± standard deviation (SD).

	Sarcopenia Group (S)				Non-Sarcopenia Group (NS)			
	T0	T1	Var (%)	*p*-Value	T0	T1	Var (%)	*p*-Value
Gender (M/F)	11/9				5/9			
Age (years)	71.5 (17.0)				68.0 (16.5)			
ICU	8/20				3/14			
BMI (kg/m^2^)	21.0 (4.2)	21.4 (4.6)	1.77%	0.930	27.3 (9.0)	27.9 (8.3)	2.23%	0.928
PhA (°)	2.6 (1.3)	3.9 (1.1)	50.00%	0.043 *	3.4 (1.3)	3.5 (0.7)	4.48%	0.350
ASM (kg)	14.1 (5.6)	15.3 (4.8)	8.51%	0.001 *	17.9 (6.3)	21.6 (7.3)	10.67%	0.009 *
M	17.1 (2.8)	17.7 (2.8)	3.51%	0.976	23.4 (7.8)	25.9 (8.5)	10.68%	0.397
F	11.4 (2.3)	11.9 (2.1)	4.39%	0.017 *	16.9 (3.8)	17.8 (5.5)	5.33%	0.046 *
HG (kg)	17.3 (5.6)	20.6 (7.2)	19.71%	0.007 *	25.7 (8.2)	26.7 (8.3)	3.89%	0.398
M	19.9 (6.1)	23.9 (6.9)	6.04%	0.079	31.2 (10.8)	33.5 (7.2)	7.32%	0.458
F	13.8 (2.9)	16.1 (2.3)	16.67%	0.035 *	22.4 (5.1)	22.2 (5.4)	−0.89%	0.983
TUG (s)	31.5 (23.5)	17.9 (15.9)	−26.83%	0.039 *	13.1 (8.1)	9.0 (3.3)	−31.29%	0.045 *

T0: at admission; T1 at discharge; ICU: how many patients were hospitalized in an Intensive Care Unit in the previous hospitalization (first number regards how many patients were admitted to intensive care in the previous hospitalization, second number regards the total number of patients in that group); BMI: body mass index; PhA: phase angle; ASM: Appendicular Skeletal Mass; HG: hand grip; TUG: time-up-and-go; Var (%) = percentage variation = (value T1 − value T0)/value T0 × 100. All values refer to mean; * statistically significant (*p* < 0.05).

**Table 2 jcm-10-05623-t002:** Respiratory parameters (SpO_2_, dyspnea) measured in different positions and after TUG at T0 and T1 in the sarcopenia group (S) and the non-sarcopenia group (NS). All data are reported as mean ± standard deviation (SD).

	Sarcopenia Group (S)				Non Sarcopenia Group (NS)			
	T0	T1	Var (%)	*p*-Value	T0	T1	Var (%)	*p*-Value
Lying								
SpO_2_%	93.0 (3.0)	95.0 (2.0)	2.15%	0.345	92.5 (4.0)	95.5 (4.3)	3.24%	0.661
Dyspnea Borg’s 0–10 scale	0.1 (0.5)	0.1 (0.1)	0%	<0.001 *	0.1 (0.1)	0.1 (0.1)	0%	<0.001 *
Sitting								
SpO_2_%	93.0 (3.5)	96.0 (1.5)	3.23%	0.004 *	94.5 (4.3)	95.5 (1.8)	1.06%	0.084
Dyspnea Borg’s 0–10 scale	0.1 (1.5)	0.1 (0.1)	0%	0.166	0.1 (0.3)	0.1 (0.1)	0%	<0.001 *
Standing								
SpO_2_%	93.0 (5.5)	96.0 (3.0)	3.23%	0.002 *	95.0 (3.0)	95.5 (3.3)	0.53%	0.578
Dyspnea Borg’s 0–10 scale	2.0 (3.0)	0.1 (0.1)	−95%	0.040 *	0.5 (2.3)	0.1 (0.1)	−80%	<0.001 *
After TUG								
SpO_2_%	91.0 (4.5)	95.0 (3.5)	4.40%	0.094	94.0 (4.5)	95.0 (1.5)	3.26%	0.821
Dyspnea Borg’s 0–10 scale	4.0 (4.4)	0.5 (2.0)	−87.50%	0.087	2.0 (3.0)	0.1 (1.5)	−95%	0.012 *

T0: at admission; T1 at discharge; SpO_2_%: oxygen saturation %; Dyspnea Borg’s 0–10 scale: perception of dyspnea; TUG: time-up-and-go; Var (%) = percentage variation = (value T1 − value T0)/value T0 × 100; * statistically significant (*p* < 0.05).

**Table 3 jcm-10-05623-t003:** Main laboratory findings at T0 and T1 in the sarcopenia group (S) and the non-sarcopenia group (NS). All data are reported as mean ± standard deviation (SD).

	Sarcopenia Group (S)				Non-Sarcopenia Group (NS)			
	T0	T1	Var (%)	*p*-Value	T0	T1	Var (%)	*p*-Value
Albumin g/dL	3.5 (0.5)	3.6 (0.6)	2.86%	0.849	4.3 (0.6)	4.4 (0.8)	2.33%	0.876
Ferritin ug/L	448.5 (367.0)	310.0 (240.0)	−30.88%	0.045 *	151.0 (197.0)	85.5 (146.5)	−25.00%	0.958
Vitamin D ug/L	13.7 (13.2)	24.7 (13.3)	80.29%	0.048 *	25.5 (13.5)	44.5 (35.9)	74.51%	0.448

T0: at admission; T1 at discharge; Var (%) = percentage variation = (value T1 − value T0)/value T0 × 100; * statistically significant (*p* < 0.05).

## Data Availability

The data presented in this study are available upon request from the corresponding author. The data are not publicly available due to their containing information that could compromise the privacy of research participants.

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
