# Peer review of "Skeletal Muscle Mass, Sarcopenia and Rehabilitation Outcomes in Post-Acute COVID-19 Patients"

_jcm, 2021, doi:10.3390/jcm10235623_

Round 1
Reviewer 1 Report
General comments:
The authors explore the prevalence of sarcopenia in COVID-19 patients and impact of skeletal muscle mass and sarcopenia at beginning of the post-acute rehabilitation phase on body composition, functional and respiratory parameters at discharge.
They reported that 34 patients were enrolled and prevalence of sarcopenia was 58%. Sarcopenic patients showed more pronounced, statistically significant improvements at discharge in body composition, respiratory function, and muscle strength as compared with non-sarcopenic patients.
This reviewer has several comments related to this study as mentioned below.
Major comments:
- Title, abstract, and introduction: Authors described that aim of this study is to investigate the relationship between “skeletal muscle mass” and “rehabilitation outcomes”. However, they divided patients into two groups, patients with sarcopenia and non-sarcopenia group, and compared several parameter between two groups. The definition of sarcopenia included low muscle mass and muscle function and/or physical performance. So, low muscle mass is not always equal “sarcopenia”. It is better to change the aim of this study from “skeletal muscle mass” to “existence of sarcopenia” or “skeletal muscle mass, muscle strength and physical performance”.
- A study of 34 subjects is likely to be too small sample size to detect differences between two groups and to draw any conclusions.
- This study showed that the prevalence of sarcopenia was 58%. Prevalence of sarcopenia in rehabilitation unit is mainly depend on hospitalization criteria in each units. Therefore, it may be difficult to extrapolate this result to other hospital or institute.
- The information of patient’s characteristics are poor. It is better to add more information in Table 1 about patient’s treatment in the ICU and comorbidity such as duration of mechanical ventilation in the ICU, duration of ICU and hospital stay, steroid or muscle relaxant use, diabetes mellitus, hypertension, chronic cardiac disease, or chronic respiratory disease, etc.
- Line 162-164: This sentence belongs to methods or statistical analysis.
- In my opinion, discussion is too long to follow it. Discussion should be shorten by deleting description of obvious facts considerably and separate into three to four section by adding line feed in the text.
- Line 289-295: The contents of these sentences are general opinion and not concur with the aim of this study. Please describe the conclusions of this study in brief at the end of this manuscript.
Minor comments:
- It is better to replace “SARS-CoV-2 patient” with “COVID-19 patients” or “patient with SARS-CoV-2 infection”.
- Line 97: Please correct ASM units from “(kg/m)” to “(kg)”.
- Table 1 to 3 would be more reader friendly if you correct such as “S to Sarcopenia group (n=20)”, “NS to Non-sarcopenia group (n=14)”.
- Line 291: Correct misspelling from “related complicances” to “related complications”
Author Response
Answer to first reviewer
Q1. Title, abstract, and introduction: Authors described that aim of this study is to investigate the relationship between “skeletal muscle mass” and “rehabilitation outcomes”. However, they divided patients into two groups, patients with sarcopenia and non-sarcopenia group, and compared several parameter between two groups. The definition of sarcopenia included low muscle mass and muscle function and/or physical performance. So, low muscle mass is not always equal “sarcopenia”. It is better to change the aim of this study from “skeletal muscle mass” to “existence of sarcopenia” or “skeletal muscle mass, muscle strength and physical performance”.
A1: we modified the aim as suggested by reviewer 1: The aim of this study was to investigate the impact of the existence of sarcopenia at admission to a post-acute COVID-19 patients rehabilitation Unit on body composition, functional and respiratory capacity at discharge.
Monitoring muscle mass and strength in post-acute SARS-CoV2 patients could be useful to predict respiratory, functional and nutritional rehabilitation outcomes.
Q2. A study of 34 subjects is likely to be too small sample size to detect differences between two groups and to draw any conclusions.
A2 we agree that the sample is too small. Considering the emergencial work condition in the hospital and consequential organizational changes during the pandemia, post-Covid 19 patients have been temporarily available in our hospital, therefore the sample size is limited. Also, allocating staff for research purposes was limited in that specific period.
Q3. This study showed that the prevalence of sarcopenia was 58%. Prevalence of sarcopenia in rehabilitation unit is mainly depend on hospitalization criteria in each units. Therefore, it may be difficult to extrapolate this result to other hospital or institute.
A3 we agree that the prevalence of sarcopenia across different rehab units depends on admission criteria. However, our aim was focused on assessing the prevalence of sarcopenia solely in patients who had just been discharged from a Covid hospital and were in need of intensive rehabilitation. Therefore, admission criteria was strictly defined by a discharge from a Covid hospital. In the discussion we reported that our results are in line with the recent reports on the prevalence of sarcopenia of the patients from intensive care units (25).
Q4 The information of patient’s characteristics are poor. It is better to add more information in Table 1 about patient’s treatment in the ICU and comorbidity such as duration of mechanical ventilation in the ICU, duration of ICU and hospital stay, steroid or muscle relaxant use, diabetes mellitus, hypertension, chronic cardiac disease, or chronic respiratory disease, etc.
A4 We added the info about patients who were admitted to ICU in Table 1. Unfortunately, information about the number of days spent in the ICU or duration of mechanical ventilation was not reported in the medical records from other hospitals, as patients were referred from Covid hospitals scattered in northern Italy. In our sample, prevalence was 17.1% for hypertension, 16.4% for cardiac-cerebrovascular disease and 9.7% for diabetes. We added that information.
Q5 Line 162-164: This sentence belongs to methods or statistical analysis.
A5 We placed the sentence in the Methods sections, Measurement subsection
Q6. In my opinion, discussion is too long to follow it. Discussion should be shorten by deleting description of obvious facts considerably and separate into three to four section by adding line feed in the text.
A6 We have modified the discussion accordingly
Q7 Line 289-295: The contents of these sentences are general opinion and not concur with the aim of this study. Please describe the conclusions of this study in brief at the end of this manuscript.
A7 We have modified the discussion accordingly
Minor comments:
- It is better to replace “SARS-CoV-2 patient” with “COVID-19 patients” or “patient with SARS-CoV-2 infection”.
We modified in COVID-19 patients
- Line 97: Please correct ASM units from “(kg/m)” to “(kg)”.
We modified as suggested
- Table 1 to 3 would be more reader friendly if you correct such as “S to Sarcopenia group (n=20)”, “NS to Non-sarcopenia group (n=14)”.
We modified as suggested
- Line 291: Correct misspelling from “related complicances” to “related complications”
Corrected
Reviewer 2 Report
The manuscript is including results of a grate work. but did not find conclusion part.
Author Response
Second Reviewer
Q1:The manuscript is including the results of a great work. but did not find conclusion part.
A1: The conclusion part is not mandatory, anyway we added that part
Reviewer 3 Report
The manuscript titled as "Skeletal muscle mass, sarcopenia and rehabilitation outcomes in post-acute SARS-COV-2 patients". This manuscript is a research study and recruited 34 post-acute SARS-COV-2 patients.
Comments and Suggestions for Authors
The paper is overall well structured despite a some comments. I have suggestion that could improve the paper:
- Introduction
- The introduction explains the most important information about the study.
- Material and methods:
- Plaese add median and quartile to continuous variables.
- For a small group should be use chi2 with Yates correction or Fisher test
- Results:
- Below the tables please add legend.
- Discussion
- In my opinion the discussion is well written.
Author Response
Answer third reviewer
- Material and methods:
- Plaese add median and quartile to continuous variables.
- For a small group should be use chi2 with Yates correction or Fisher test
We changed as suggested
- Results:
- Below the tables please add legend.
We added legends in all tables.
Reviewer 4 Report
Good article since the subject is novel and as of higher importance due the pandemic situation and the lack of knowledge about long covid
However i have some concerns about the objectiv and methodology
OBJECTIV: first aim ...was to investigate the impact of skeletal muscle mass ..." - Not clear! the impact of SM on what?
The second aim (investigate the prevalence of sarcopenia...) does not need an intervention time and surelly need an higher sample
METHODOLOGY
Authors have refered the age by means (mean +/- SD). That not permit to perceive the amplitude of age of participants. And that is of main interest since TUG is only for 65years and plus. The values presented could not show this and this could be a huge error! So please, correct that and present the amplitude of age of participants
Also values considered for cutt-off in TUG are not clear and not explained well. Cutt-off considered for sarcopenia is 20seg and values considered of reference for risk of falls is 13,5 and that are values recognized and validated. However authors chosed the value of 12seg obtained in a study of 2003, older than the more recent studies of validation. Please explain that.
Again TUG, in my opinion is not very well explained since authors claimed that is for "physical performance" (???) but TUG is studied and validated for mobility, balance, walking ability and fall risk in older adults. Please explain and correct.
Another concern that i have is about muscle mass. Is great that a short period as 28 days could statistically increase muscle mass. Major studies about relationship between training, muscle mass and strength, do not show that. The increase in muscle mass is slower than the increase in muscle strength and ...in 28 days??? Could you discuss this subject? Handgrip strength has increased in sarcopenic but not in NS? But muscle mass has increased? Not make real sense to me.
Why have authors decided not to assess lower muscle strength? That was os paramount importance in my opinion.
Great assessment of body composition with bioimpedance but why have authors ignored fat mass? of course for sarcopenia , fat does not matter but could help to explain why have BMI changed between T0 and T1, althougth not significantly.
.
Author Response
Q1: first aim ...was to investigate the impact of skeletal muscle mass ..." - Not clear! the impact of SM on what?
A1: we modified the aim as suggested also by reviewer 1: The aim of this study was to investigate the impact of the existence of sarcopenia at admission to a post-acute SARS-COV-2 rehabilitation Unit on body composition, functional and respiratory capacity at discharge. Our results emphasize that assessing and monitoring muscle mass and strength in post-acute SARS-CoV2 patients could be useful to predict respiratory, functional and nutritional rehabilitation outcomes.
Q2 The second aim (investigate the prevalence of sarcopenia...) does not need an intervention time and surelly need an higher sample
A2 we rephrased it into the single aforementioned aim
METHODOLOGY
Q3 Authors have referred the age by means (mean +/- SD). That not permit to perceive the amplitude of age of participants. And that is of main interest since TUG is only for 65years and plus. The values presented could not show this and this could be a huge error! So please, correct that and present the amplitude of age of participants
A3 We divided in two groups (sarcopenic and non-sarcopenic) and for each group we defined 6 age ranges.
According EWGSOP2 (A.J. Cruz-Jentoft et al. 2019) (https://www.ncbi.nlm.nih.gov/pmc/articles/PMC6322506/pdf/afy169.pdf )TUG performance test is to be used not exclusively for patients over 65 (Table 3).
Q4 Also values considered for cut-off in TUG are not clear and not explained well. Cutt-off considered for sarcopenia is 20sec and values considered of reference for risk of falls is 13,5 and that are values recognized and validated. However authors chosed the value of 12seg obtained in a study of 2003, older than the more recent studies of validation. Please explain that.
A4 The 2003 article below, suggesting a 20-sec cut-off value, has been cited as number 20 in our references. This is the same article reported in the guidelines of Sarcopenia: revised European consensus on definition and diagnosis when discussing about TUG.
Bischoff, H.A.; Stähelin, H.B.; Monsch, A.U.; Iversen, M.D.; Weyh, A.; von Dechend, M.; Akos, R.; Conzelmann, M.; Dick, W.; Theiler, R. Identifying a cut-off point for normal mobility: A comparison of the timed “up and go” test in community-dwelling and institutionalised elderly women. Age and Ageing 2003, doi:10.1093/ageing/32.3.315.
In the 2019 guidelines. 20 seconds are the cut off for a definition of severe sarcopenia. We acknowledge that risk of falling is defined by a 13.5 sec cut-off, as reported in the 2014 paper, however, the cut-off for sarcopenia remains at 20 sec.
Q5 Again TUG, in my opinion is not very well explained since authors claimed that is for "physical performance" (???) but TUG is studied and validated for mobility, balance, walking ability and fall risk in older adults. Please explain and correct.
A5 TUG is considered part of the physical performance tests (see, among others, Cruz-Jentoft et al. 2019, https://www.ncbi.nlm.nih.gov/pmc/articles/PMC6322506/pdf/afy169.pdf )
Q6 Another concern that i have is about muscle mass. Is great that a short period as 28 days could statistically increase muscle mass. Major studies about relationship between training, muscle mass and strength, do not show that. The increase in muscle mass is slower than the increase in muscle strength and ...in 28 days??? Could you discuss this subject? Handgrip strength has increased in sarcopenic but not in NS? But muscle mass has increased? Not make real sense to me.
A6 Training against resistance can induce a statistically significant hypertrophic response (cross-sectional area) in as little as 20 days (https://journals.physiology.org/doi/full/10.1152/japplphysiol.00789.2006). In any case, sarcopenic patients increased both lean mass and strength. Also, we reported in the discussion:
“Also, it can be speculated that patients in S, starting from a more severely de-conditioned status, benefited more from the rehabilitation intervention than NS. It is known that the relationship between strength and function is curvilinear, suggesting that other factors than muscle function contribute to function, still muscle strength is considered the most important (38). Hand grip strength is a proxy for global strength measurements and it has been used to identify sarcopenia in clinical and community health settings (39). Deconditioned, frail patients below the threshold of independence more likely achieve clinically significant functional gain in basic activities of daily living, as compared to patients who are already functionally closer to independence, namely the NS”
Q7 Why have authors decided not to assess lower muscle strength? That was os paramount importance in my opinion.
A7 Handgrip is universally used as a proxy of global strength and has been used to identify sarcopenia in clinical and community health settings. Also, operators are familiar with the routine execution of the test, whereas lower limb strength testing is less common. But more importantly, our patients were not able at the baseline testing to execute a chair stand test, reported in Cruz-Jentoft et al. 2019, where the patient is asked to cross his arms and not use other aids.
Q8 Great assessment of body composition with bioimpedance but why have authors ignored fat mass? of course for sarcopenia , fat does not matter but could help to explain why have BMI changed between T0 and T1, althougth not significantly.
A8 In patients, changes in fat mass were not statistically significant in contrast to the reported ASM. The change in BMI can be explained by an increase in total body water. Patients were better hydrated. Thus, it is possible that the estimation of muscle mass by BIA may have been influenced by the increase in total body water in the patients. Last but not least, the BIA cannot determine distribution of fat, which is calculated as a difference between body mass and lean mass (Assessment of Body Composition in Athletes: A Narrative Review of Available Methods with Special Reference to Quantitative and Qualitative Bioimpedance Analysis. MDPI/NUTRIENTS 2021 by Campa F et al). The estimate of lean mass depends on the amount of total water; as total body water varies, the estimated lean mass may also vary.
Round 2
Reviewer 1 Report
Thank you for your effort and time spent on this revision. The authors have answered my queries.
Minor comments:
- Table 1-3: What is the blank space between ”T1” and "p-value" in revised version? Absolute difference (%) ? If so, please fill in absolute difference (%) (95% CI) . In table 2, please add explanation of "var (%)".
- Table 1-3: Are all values mean (SD)? Please describe it.
- Table 1: It may be better to delete number of patients for age range from table 1 because it seems too busy. In addition, this reviewer can not understand ICU "8/20" and " 3/14". Please describe it.
- Table 1-3: All values and group names in the tables should be centered.
- Table 3: Please correct from #Vitamina D" to " Vitamin D".
- Table 1: ASM: Please align F and its values.
- Line 234-235: Please correct to " inflammatory state ferritin 457.17 (290.71) vs. 201.62 (194.99) and more . . . "
- Line 304: Please delete " (9)".
Author Response
Answer to Reviewer
Q1: Table 1-3: What is the blank space between ”T1” and "p-value" in revised version? Absolute difference (%) ? If so, please fill in absolute difference (%) (95% CI) . In table 2, please add explanation of "var (%)".
A1: Var (%) = Percentage Variation =(value T1-value T0)/value, thank you for the correction.
Q2: Table 1-3: Are all values mean (SD)? Please describe it.
A2: Yes, all data are reported as mean ±standard deviation (SD), we added the comment in all tables.
Q3: Table 1: It may be better to delete number of patients for age range from table 1 because it seems too busy. In addition, this reviewer can not understand ICU "8/20" and " 3/14". Please describe it.
A3: We removed the age range as suggested and we reported a description in the table. first number regards how many patients were admitted to intensive care in the previous hospitalization, second number regards the total number of patients in that group
Q4 Table 1-3: All values and group names in the tables should be centered.
A4: We changed as suggested
Q5 Table 3: Please correct from #Vitamina D" to " Vitamin D".
A5: We changed as suggested
Q6: Table 1: ASM: Please align F and its values.
A6: We changed as suggested
Q7 Line 234-235: Please correct to " inflammatory state ferritin 457.17 (290.71) vs. 201.62 (194.99) and more . . . "
A7: We changed as suggested
Q8 Line 304: Please delete " (9)".
A8: We changed as suggested